# Development of Biodegradable PLA/PBAT-Based Filaments for Fertilizer Release for Agricultural Applications

**DOI:** 10.3390/ma15196764

**Published:** 2022-09-29

**Authors:** Thyago Camelo Pereira da Silva, Allef Gabriel da Silva Fortes, Iago Rodrigues de Abreu, Laura Hecker de Carvalho, Yeda Medeiros Bastos de Almeida, Tatianny Soares Alves, Renata Barbosa

**Affiliations:** 1Graduate Program in Materials Science and Engineering, Technology Center, Federal University of Piauí, Teresina 64049-550, PI, Brazil; 2Center Science and Technology, Graduate Program in Materials Science and Engineering, Federal University of Campina Grande, Campina Grande 58428-830, PB, Brazil; 3Center for Technology and Geosciences, Graduate Program in Chemical Engineering, Federal University of Pernambuco, Recife 50670-901, PE, Brazil

**Keywords:** controlled release, nanocomposites, 3D printing, biodegradable

## Abstract

The aim of this work was to produce filaments of PLA/PBAT and NPK fertilizer adsorbed on organophilized bentonite intended for application in the prototyping of biodegradable agricultural artifacts in 3D printing, using the Fused Deposition Modeling (FDM) technique. This is the first time that we have reported this composite for a 3D printing approach. Systems containing PLA/PBAT, organobentonite and NPK were initially processed in an internal mixer and later extruded as filaments in a single-screw extruder. The prototypes were printed by FDM. Structural, morphological and thermal properties, as well as NPK releasing, were investigated. The results suggest that exfoliated and/or intercalated nanocomposites were obtained by the organoclay addition to the PLA/PBAT blend. The morphological analysis revealed a good surface quality of the impressions. Systems containing organobentonite released approximately 22% less fertilizer in 24 h compared to the systems without organobentonite. This difference is due to the higher concentration of nanoparticles that generate more barriers to the diffusion of NPK. The release data for these systems had a better fit to the kinetic model of Korsmeyer-Peppas. Thus, studied filaments have the potential to retard the release of fertilizer and are suitable for further development of structures for agricultural applications by FDM.

## 1. Introduction

The world agricultural industry uses significant amounts of mineral fertilizers. The world agricultural demand for NPK (mixed mineral fertilizer based on nitrogen—N, phosphorus—P and potassium—K) forecast for 2022 is of the order of 200 thousand tons and an expected CAGR of 5.0% by 2028 [1,2]. The continent of greatest demand is Asia, due to factors such as economies focused on agriculture, large areas that allow cultivation and high population density, which requires greater productivity of agricultural crops. For example, China and India rank second and third in urea consumption, respectively. However, this high demand comes at a cost, due to the unregulated application of these types of fertilizers, nutrient loss can occur, as indicated by Singh et al. (2021), where India had a negative balance of approximately 8 to 10 MT year^−1^, and the situation is expected to worsen, reaching 15 MT year^−1^ [2,3,4]. Furthermore, traditional NPK-type fertilizers have significant polluting potential, resulting from volatilization losses which generate atmospheric pollution and leaching of oxidized nutrients, which in turn contaminate aquifers. The use of fertilizers with controlled or slow release (CRF) stand out among the recent solutions studied to mitigate these impacts [5,6,7].

CRFs reduce losses by delivering nutrients more efficiently as they are designed to release active fertilizer compounds in a delayed or controlled manner, in line with the specific nutrient uptake needs of plants [8]. In general, CRFs are produced from the encapsulation of agrochemical macro- and micronutrients, such as urea or NPK, by structures that control the diffusion of water, gases and the release of the active agent [5,9].

Biodegradable polymers stand out among the materials used for the encapsulation or in the distribution of the active agent dispersed in it. The use of biodegradable polymer matrices, such as poly(hydroxybutyrate) (PHB) [10,11,12,13], poly(lactic acid) (PLA) [14], starch [15,16,17], lignin [18], carboxymethyl cellulose [19] and chitosan [20,21], is doubly advantageous because during their microbial degradation, nutrients are released with greater control in the environment, minimizing losses and reducing the accumulation of residues in the soil in a sustainable way.

Allied to the use of biodegradable plastics for sustainability, industry 4.0 presents itself as a crucial tool in increasing productivity and efficiency in the planning and control of products in several segments, with agricultural development being the most recent. Additive manufacturing, one of the pillars of this new industry, comprises a low-cost, efficient, easy-to-operate production technique which provides a wide variety of materials that can be used in versatile applications [22,23,24] ranging from biomedical, automotive and aerospace applications as well as prototyping industries in general [25].

In recent years, 3D printing has been used in drug delivery applications, such as in the manufacture of tablets and in the manufacture of dressings containing active pharmaceutical substances [26]. This technology potentially can be applied in the development of agricultural artifacts, traditionally produced by other processing means, as well as enable the production of systems with biodegradable polymer matrices to control the release of fertilizers.

The association of biodegradable matrices with inorganic nanoparticles is an interesting alternative to improve the performance of CRFs. A polymer matrix with potential for application in controlled release and association with nanoparticles is Ecovio^®^, a commercial biodegradable thermoplastic blend produced by BASF. Ecovio^®^ is composed of PLA and poly(butylene adipate co-terephthalate) (PBAT) (Ecoflex^®^) and is a material with mechanical properties similar to those of traditional polymers used for agricultural applications [27,28]. Organophilic bentonite clays stand out among the inorganic nanoparticles applicable to the adsorption of fertilizers and their CRF biodegradable nanocomposites have greater matrix stability which allow for the slow release of agricultural compounds [5,29]. In addition to controlling drug release, nanoparticles can provide improvements in general properties of the composite, such as tensile strength, elastic modulus, flexural strength, flexural modulus [30,31], barrier properties [32] and thermal stability [33]. These improvements expand the application possibilities of these systems and may even result in materials with suitable characteristics for application in the production of planting artifacts with high added value, such as fertilizer sticks, biodegradable tubes and soilless agriculture substrates able to release nutrients.

Therefore, the present work aims to develop PLA/PBAT nanocomposite filaments, in which the organophilic bentonite clay filler is incorporated with NPK fertilizer, to be used in Fusion and Deposition Modeling (FDM) prototyping of biodegradable agricultural artifacts capable of releasing nutrients. To the best of our knowledge, this is the first time that a sustainable bio-based nanocomposite filament for a 3D printing approach capable of releasing fertilizer with the potential to be applied in several innovative planting techniques is reported. 

## 2. Materials and Methods

### 2.1. Materials

The PLA/PBAT blend, commercially known as Ecovio^®^, grade F2224, supplied by BASF S/A (Camaçari, BA, Brazil) with a density of 1.24–1.26 g/cm³ and composed of 45% PLA and 55% PBAT, was used as polymeric matrix. The organophilized montmorillonite clay used as a filler and fertilizer carrier was commercial bentonite Cloisite 20A^®^ (Southern Clay Products Inc., Gonzales, TX, USA), particle size < 10 μm, basal spacing (XRD, d001): 2.7 nm. As fertilizer, a mixed mineral compound was used, NPK 4-14-8 with 4% monoammonium phosphate or ammonium dihydrogen phosphate (MAP) (NH_4_H_2_PO_4_), 14% simple superphosphate (P_2_O_5_ + Ca + S) and 8% potassium chloride (KCl), produced by Fertilizantes Tocantins S/A (São Luís, MA, Brazil).

### 2.2. Preparation of Composites and Production of Filaments

A schematic illustration of preparation of composites, production of filaments and 3D printing of the samples is shown in Figure 1. Concentrates were prepared in a Haake™ Rheomix 3000 (Thermo Fisher Scientific, Waltham, MA, USA) internal mixer fitted with roller-type rotors coupled to a torque rheometer operating at 180 °C and 60 rpm, for 8 min (Figure 1a). During processing, after the melting of Ecovio^®^, identified by the stabilization of torque after an average time of 4 min, NPK and bentonite clay were simultaneously added. Subsequently, the concentrates were diluted to obtain the filaments for 3D printing in an AX-16 (AX Plásticos, Diadema, SP, Brazil) single-screw extruder, with 16-millimeter screw diameter (L/D = 26), operating with a temperature profile of 150, 155 and 155 °C in its three heating zones and screw speed of 30 rpm (Figure 1b). Filament compositions are summarized in Table 1.

Filament dimensions were controlled in a FTR1 pulling system (Filmaq3D, Curitiba, PR, Brazil) and a winding tool (FE1, Filmaq3D, Curitiba, PR, Brazil) (Figure 1b). Pulling parameters were adjusted by changing the rotation speed of the winder roll, initially set at 35 rpm, in order to produce filaments with diameters close to 1.75 mm. Filament diameters were measured with a Digital Caliper^®^ (0–150 mm) and presented in Table 2.

### 2.3. Prototype Printing 

Thin-walled tubular prismatic specimens (20 × 20 × 20 × 1 mm) were obtained with a 3D printer Da Vinci^®^ 1.0 Pro. (XYZprinting Inc., New Taipei City, Taiwan) with a 0.2-millimeter print nozzle (Figure 1c). The printing parameters defined in the preliminary tests to determine the ideal processing conditions of the filaments by FDM are summarized in Table 3.

### 2.4. Characterization

The morphology of the printed surfaces was analyzed by scanning electron microscopy (SEM) in a TESCAN VEGA 3 instrument (TESCAN ORSAY HOLDING, a.s., Brno, Kohoutovice, Czech Republic). X-ray diffractometry (XRD) was performed on samples from the walls of the printed specimens on a Shimadzu X-ray diffractometer (LABX-XDR 600, Shimadzu, Kyoto, Japan), operating in the angular range from 2° to 40° (2θ), with a speed of 2°/min and a power of 40 kV/30 mA, with incident radiation Cu-Kα (λ = 1.5406 Å). Thermogravimetric analyses (TGA) were carried out in an equipment Mettler-Toledo Ind. e Com. (Barueri, SP, Brazil), under nitrogen atmosphere with a flow of 50 mL·min^−1^ from 0 to 600 °C and a heating rate of 10 °C·min^−1^.

### 2.5. Release Rate in Water and Release Kinetics

The release profile of NPK salts from the filaments was indirectly determined by measuring the total concentration of electrolytes in solution by ionic conductivity in deionized water using a portable conductivity meter (MB-11P, Marte Científica, Santa Rita do Sapucaí, MG, Brazil), according to the procedure described by Daitx et al. (2019) [10] and Scaffaro, Citarrella and Gulino (2022) [34]. Average values of triplicate measurements, performed at room temperature, are reported. The NPK content released was calculated with the following Equation (1):(1)$$A_{r}=\frac{V_{s}\sum_{1}^{n-1}C_{i}{+V}_{0}C_{n}}{V_{0}C_{t}}\;\times 100$$
where A*_r_* is the percentage of NPK released (%), V*_s_* is the sample volume (mL), C*_i_* and C*_n_* are the conductivities of the samples (μS cm^−1^) at time *i* through *n*, V_0_ is the initial test volume (mL) and C*_t_* is the theoretical total conductivity (μS cm^−1^). The theoretical total conductivity value was based on the free NPK content released in deionized water. 

The rate release of the systems and their physical kinetics mechanism were determined by comparing the release data with zero-order and first-order mathematical models and the Higuchi and Korsmeyer-Peppas models [35,36,37].

## 3. Results and Discussion

### 3.1. Morphological Analysis

The morphological analysis of the composites was performed by SEM of the external surfaces of the printed specimens. Figure 2 shows that the external surfaces of the matrix and composites taken at 50× magnification have good quality as there was no formation of voids and agglomerates. There is no apparent porosity and there was no substantial change in the interlayer adhesion with the addition of NPK and bentonite. For the 200× magnifications, in Figure 2b*–d*, some agglomerates are observed, characteristic of the clay particulate fillers and NPK fertilizer used [38].

Figure 2e*,f* show a detached fillet in the micrograph, which is attributed to poor adhesion during filament layer deposition as a result of incomplete melting and diffusion of vicinal strands. These discontinuities and/or voids can be associated with the deposition of the filament in layers in the formation of the part. During deposition, gaps may be partially filled (pores) due to incomplete diffusion into adjacent strands in some parts, and slight agglomeration or saturation of the filling may be observed [22,39].

The micrographs indicate that there was a good filler/matrix interfacial interaction. It is believed that the composites surface roughness may be associated to evaporation of water or other substances present in inorganic components, generally associated with their ability to interact with ambient humidity [10,40]. All samples exhibited approximately 200-micrometer thick layers, conforming to the layer height standard (0.2 mm) set at printing. Despite some observed discontinuities, all samples showed geometric regularity and good visual finish, indicating that the filaments produced are suitable for use in 3D printing.

### 3.2. XRD Analisys

XRD spectrum patterns for the organophilic bentonite Cloisite 20A^®^, fertilizer—NPK and composites with Ecovio^®^ polymer blend—are shown in Figure 3a. Cloisite 20A^®^ presents two characteristic peaks, one referring to 2θ diffraction ≈ 3.5°, associated with the (001) plane with interplanar spacing of 2.48 nm and another to a second-order peak (reflection) of the (002) plane at 2θ ≈ 7.2° with a 1.23-nanometer spacing, both calculated by Bragg’s law (2d sin θ = n λ) (Figure 3a) [41,42].

The NPK spectrum shows specific peaks for urea at 2θ = 24°, 25°, 28°, 31°, 34° and 38° [43,44], in addition to peaks attributed to ammonium dihydrogen phosphate at 2θ = 16.8°, 23.7°, 29.4°, 33.8° and 38° and potassium dihydrogen phosphate at 2θ = 16.8°, 23.7°, 33.8° and 38° (Figure 3a) [44,45].

By analyzing Figure 3a, it is possible to observe that Ecovio^®^ (Ecov) presents characteristics of both polymers that constitute it, so the graphic shows a large amorphous halo and diffraction peaks at 16.2°, 17.4°, 20.4°, 23.1° and 24.9°. The literature reports that the crystalline phase of the PLA presents the diffraction angles in approximately 16.2° and 17.4°, referring to the planes (110) and (203) [46,47]. However, Arruda et al. (2015) [48] related the peaks in values close to those found in this study to basal reflections of the planes (010), (111), (100) and (111), characteristic of the PBAT crystalline phase that eventually overlapped the crystalline phase of the PLA. Similarly, Wu, et al. (2012) [49] found close values for the crystalline phase of PBAT.

There was an increase in the interplanar distances for the nanocomposites containing bentonite (Figure 3b), as a shift of the characteristic peak of Cloisite 20A^®^ from 2θ values of 3.5° to close to 2,6° for d_001_ and of the reflection peak from 7.20° to 4.9° is observed. Interplanar distances of nanocomposites Ecov/3Bent, Ecov/3Bent/3NPK and Ecov/6Bent/3NPK of 1.78 nm, 1.79 nm and of 1.81 nm, respectively, were achieved in relation to the second-order peak previously highlighted.

The shift of the peaks at 3.5° and 7.20° to lower angles and the interplanar distance increasing of the d_001_ plane suggest intercalation in the nanocomposites with the addition of clay [28,41,50]. This result demonstrates the potential for delay in the release of NPK fertilizer, which can be corroborated by the reduction in the release rate for systems containing Cloisite 20A^®^ bentonite, either via barrier mechanism to the diffusion of chemical species in the matrix or by its incorporation into the structure interspersed within the clay. Similar results were observed in the literature [51,52].

The concentration of NPK used in this study was not enough to cause phase changes in the Ecovio^®^ blend nor to show the diffraction peaks of the same. Only peaks belonging to the semi-crystalline phase of the blend, attributed to poly (butylene adipate co-terephthalate) (PBAT), were observed in the systems Ecov/3NPK, Ecov/3Bent/3NPK and Ecov/6Bent/3NPK. 

De Lima Souza, Chiaregato and Faez (2018) [53], however, in their studies on PHB/Clay/NPK, stated that at concentrations of 10% of NPK, the intensity of the Polyhydroxybutyrate (PHB) peaks decreased due to NPK dilution. The authors also identified that the peak attributed to the basal spacing of montmorillonite at 2θ = 5° disappeared in PHB/Clay/Fertilizer systems, indicating that an exfoliated structure was obtained.

### 3.3. Thermal Analysis

The thermal stability of the composites was analyzed by thermogravimetric measurements, as shown in the TGA and Derivative Thermogravimetry (DTG) thermograms in Figure 4.

TGA and DTG curves of the Ecovio^®^ blend show the initial temperature (Ti) and the final degradation temperature (Tf) for both polymers in the blend. The initial degradation temperature of PLA and PBAT are, respectively, 300 °C and 361 °C, and their final degradation temperatures are 361 °C and 450 °C. Peaks at approximately 360 °C refer to the decomposition of PLA, and the second event with a peak at approximately 401 °C refers to the thermal degradation of PBAT. The two distinct events can be attributed to poor thermodynamic compatibility between the two polymers, as reported in the literature [27,54,55,56].

The data show that PBAT has greater thermal stability than PLA. This is due to the fact that PBAT has a benzene ring in its molecular structure, which suppresses the movement of the polymer chain, resulting in higher thermal stability [57]. Degradation of PLA takes place by main chain scission and the formation of products such as cyclic oligomer, lactide and carbon monoxide [58].

The Ecov/3Bent system shows a shift to a lower degradation temperature in the PLA region. Similar behavior was reported by Taleb et al. (2021) [59] on PLA nanocomposites reinforced with modified nanoclays. The authors related the accelerated decomposition involving the clay for compositions of up to 3% to the hydrolysis reaction arising from small fractions of water or impurities that may exist in the montmorillonite, catalyzed by hydroxyl groups. As described by Alves et al. (2020) [60], who related degradation behaviors for PLA nanocomposites with different types of clays, including Cloisite 20A^®^, there is a relationship between the organic and inorganic portions through the hydrolysis generated by the fractions of water and impurities found in clay mineral, generating a catalytic effect and/or aminolysis associated with the ammonium surfactant. However, Tan, He and Qu (2019) [61] showed different results in their study with PLA/PBS/OMMT nanocomposites. The authors demonstrated that with uniformly dispersed, intercalated and exfoliated clay, in compositions of up to 5% clay, the thermal stability increased. This result was attributed to the good dispersion of clay in the matrix. In this study, in compositions above 5%, there was a decrease in thermal stability, which, according to the authors, is due to excess oxygen from the hydroxyl group of the surfactant in the clay or poor dispersion, which was not observed in their work.

In our work, no significant variation in the thermal stability of the blend was observed for the other clay and/or fertilizer compositions (Ecov/3NPK, Ecov/6Bent, Ecov/3Bent/3NPK, Ecov/6Bent/3NPK), showing that there was no expressive variation in thermal stability compared to pure Ecovio^®^.

### 3.4. NPK Release

The average percentages of active compounds released by the filaments are shown in Figure 5. All systems exhibited considerable attenuation in the release of active compounds compared to free NPK. In up to 24 h of testing, the systems average cumulative release rate of NPK was, respectively, 21.00% and 19.67% for Ecov/3Bent/3NPK and Ecov/6Bent/3NPK. In this initial period, systems containing bentonite released about 22% less fertilizer than the Ecov/3NPK system. This difference is probably due to the higher concentration of nanoparticles that generate more barriers to the diffusion of NPK chemical species [62,63]. The Ecov/6Bent/3NPK system showed the greatest potential for release control with an average cumulative release rate of 81.85% after 120 h, while in the same period, the Ecov/3NPK system released 100% of the NPK.

### 3.5. Release Kinetics

Figure 6 shows the NPK release profiles using different mathematical models described in the literature [64]. The data show that the release curves fitted better to the semi-empirical models of Higuchi and Korsmeyer-Peppas due the highest correlation coefficient (R^2^) values. The Higuchi model describes the release of the active compound homogeneously dispersed in a uniform matrix, which behaves as a diffusion medium [65,66].

The correlation coefficients for the Korsmeyer-Peppas model were 0.8998(Ecov/3NPK), 0.8386(Ecov/3Bent/3NPK) and 0.7853(Ecov/6Bent/3NPK). This model is used when there is a possibility that more than one release mechanism is involved and the characterization of the release kinetics is performed based on the value of n (release exponent) [35,36,44]. This range of n values, between 0.5 and 1.0, characterizes non-Fickian anomalous transport. In this case, the release kinetics are conducted by swelling and diffusive phenomena [67]. The sorption and transport of molecules are directly affected by the existence of microcavities present in the matrix and the diffusion and relaxation rates of the polymer chains are similar [68,69]. 

As previously shown in in Figure 5, the release profile of NPK from all the samples exhibited a pronounced initial release phase until 48 h of immersion time. This profile is probably due to burst release of fertilizer available in systems surface, which occurs immediately (because of NPK inorganic salts instantaneous dissolution in water), followed by a subsequent release governed by diffusive phenomena [34]. For Ecov/3NPK, a value of n = 0.52 and fast kinetics were observed compared to systems with organobentonite (Figure 6d). This value close to 0.5 indicates that kinetics of this system were mostly driven by diffusive phenomena. For Ecov/3Bent/3NPK and Ecov/6Bent/3NPK, after 48 h, the slope of the curves changed and exhibited a remarkable slower release rate. The release exponents for these systems were, respectively, n = 0.68 e n = 0.58, indicating that the kinetics were probably governed by diffusion and swelling phenomena associated with the barrier effect generated by organobentonite [34,70,71].

## 4. Conclusions

In this study, we produce filaments of PLA/PBAT and NPK fertilizer adsorbed on organophilized bentonite intended for application in the prototyping of biodegradable agricultural artifacts in 3D printing using the FDM technique. The nanocomposite filaments produced have diameter ranges suitable for application in commercial 3D printers. Optimal parameters were defined for printing the filaments, which resulted in pieces with dimensional regularity. Scanning electron microscopy showed that a good surface quality of the printed specimens was achieved. Thermogravimetric analysis showed that the thermal stability of the matrix was not significantly altered by nanofiller and/or fertilizer incorporation. Organoclay addition was effective in increasing fertilizer release period. Especially, Ecov/6Bent/3NPK has the greatest potential for release control with an average cumulative release rate of 81.85% after 120 h. Finally, the filaments produced have the potential for slow or controlled release of fertilizer and are very promising to be applied in FDM in innovative planting techniques that can reduce the release of active compounds into the environment and mitigate environmental impacts. For future studies, the comparison of different types of organobentonite can be considered in order to evaluate the influence of surfactant (ionic and nonionic) on the adsorption of chemical species from fertilizers and on release performance. A continuous process based on a twin-screw extruder can also be considered in order to simplify filament production, and future research will be pursued to determine the ecotoxicity of nanocomposites.

## Figures and Tables

**Figure 1 materials-15-06764-f001:**
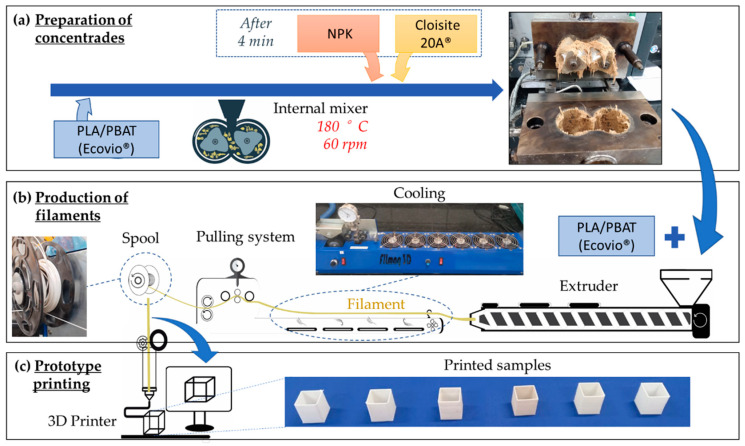
Schematic illustration of: (**a**) preparation of concentrates; (**b**) production of filament; (**c**) prototype printing.

**Figure 2 materials-15-06764-f002:**
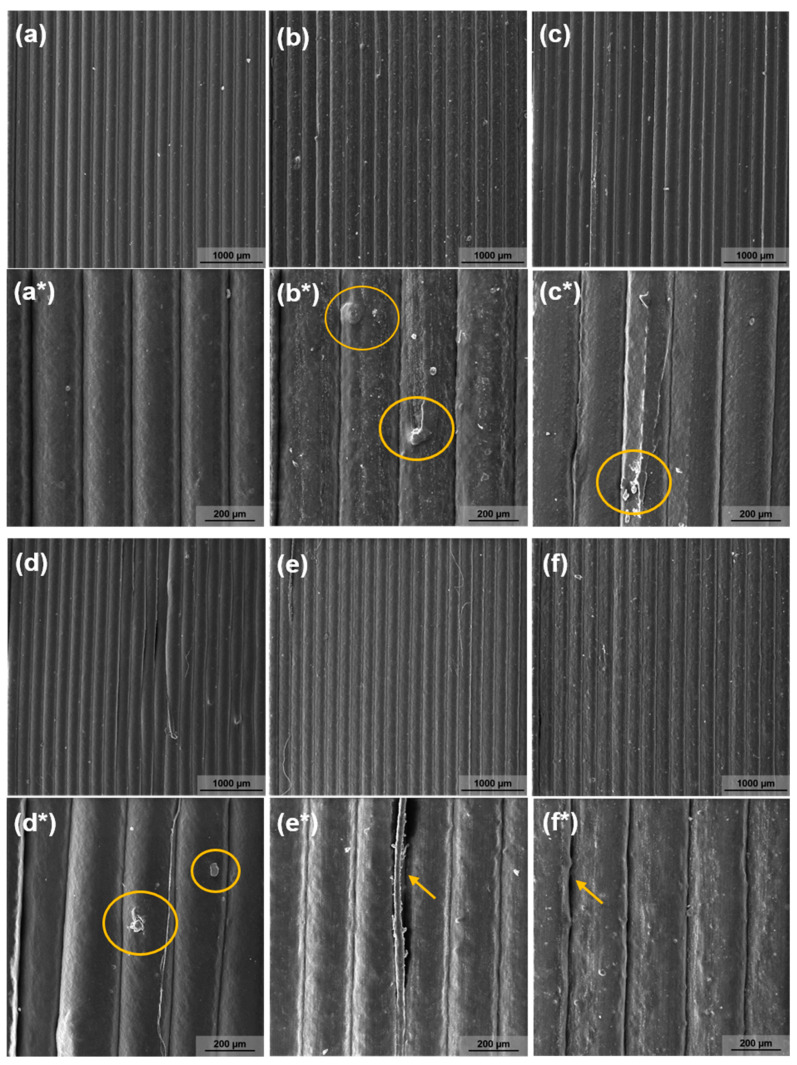
SEM micrographs of the external surface of a 3D printed sample for different compositions and magnifications: 50× [(**a**) Ecov, (**b**) Ecov/3NPK, (**c**) Ecov/3bent, (**d**) Ecov/6bent, (**e**) Ecov/3Bent/3NPK and (**f**) Ecov/6bent/3NPK]; 200× [(**a***) Ecov, (**b***) Ecov/3NPK, (**c***) Ecov/3bent, (**d***) Ecov/6bent, (**e***) Ecov/3Bent/3NPK and (**f***) Ecov/6bent/3NPK].

**Figure 3 materials-15-06764-f003:**
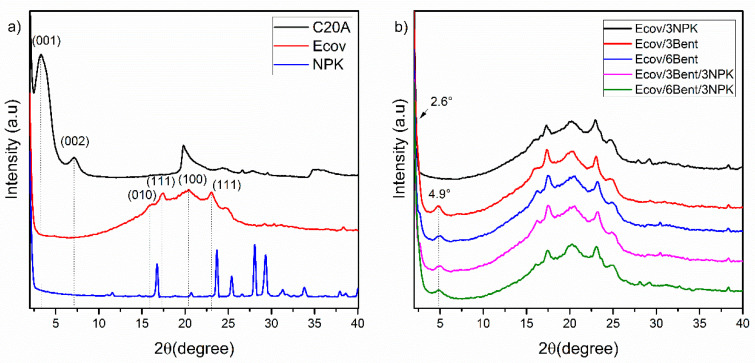
XRD of: (**a**) Cloisite 20A^®^ (C20A), NPK and neat Ecovio^®^ (Ecov); (**b**) XRD of: Ecov/3NPK, Ecov/3Bent, Ecov/6Bent, Ecov/3Bent/3NPK and Ecov/6Bent/3NPK.

**Figure 4 materials-15-06764-f004:**
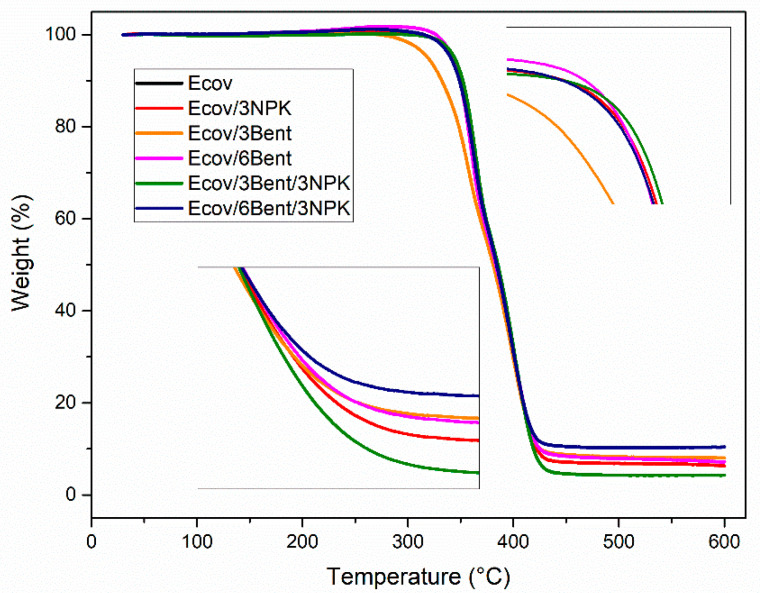
TGA of the systems: Ecov; Ecov/3NPK; Ecov/3Bent; Ecov/6Bent; Ecov/3Bent/3NPK; Ecov/6Bent/3NPK.

**Figure 5 materials-15-06764-f005:**
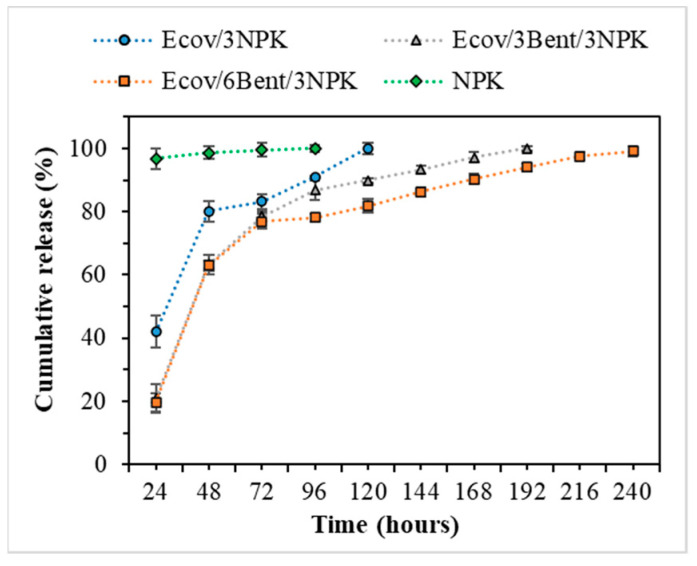
Average NPK release percentage.

**Figure 6 materials-15-06764-f006:**
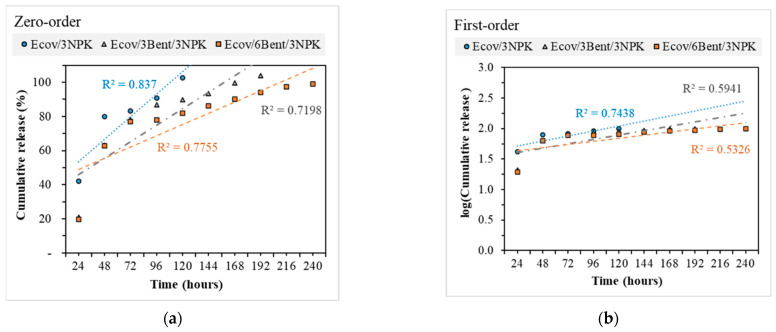
Data fitting to (**a**) zero-order; (**b**) first-order; (**c**) Higuchi; (**d**) Korsmeyer-Peppas kinetic models for NPK release by the different composites investigated.

**Table 1 materials-15-06764-t001:** Filament compositions.

Identification	Bentonite (% *w*/*w*)	NPK (% *w*/*w*)
Ecov	-	-
Ecov/3NPK	-	3
Ecov/3Bent	3	-
Ecov/6Bent	6	-
Ecov/3Bent/3NPK	3	3
Ecov/6Bent/3NPK	6	3

Legend: Ecov = Ecovio^®^, Bent = Cloisite 20A^®^, organophilic bentonite.

**Table 2 materials-15-06764-t002:** Filament diameters.

Identification	Diameter (mm)
Ecov	1.75 ± 0.05
Ecov/3NPK	1.74 ± 0.07
Ecov/3Bent	1.76 ± 0.06
Ecov/6Bent	1.75 ± 0.08
Ecov/3Bent/3NPK	1.77 ± 0.07
Ecov/6Bent/3NPK	1.77 ± 0.08

**Table 3 materials-15-06764-t003:** Filament printing parameters.

Filling Density	No filling
Higher and Lower Layers	0
Number of Walls	2
Printing Speed	30 mm/s
Travelling Speed	50 mm/s
Printing Temperature	170 °C
Print Table Temperature	45 °C
Filament Diameter	1.75 ± 0.10 mm

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
