# Peer review of "Development of Biodegradable PLA/PBAT-Based Filaments for Fertilizer Release for Agricultural Applications"

_materials, 2022, doi:10.3390/ma15196764_

Round 1

Reviewer 1 Report

In this manuscript, da Silva et al. developed biodegradable PLA/PBAT-based filaments for fertilizer release for agricultural applications. Structural, morphological, and thermal properties and the NPK releasing were evaluated. While the novelty of the entire work is insufficient and the research contents are not enough. In particular, the PLA/PBAT-based filaments proposed by the authors are quite common and cannot contribute to the progress of high-performance nanocomposite filaments. Therefore, I think the current paper is not suitable for the Materials.

Author Response

Dear Editors,

Thank you for the opportunity to revise our manuscript "Development of biodegradable PLA/PBAT-based filaments for fertilizer release for agricultural applications". We have revised the manuscript in line with the reviewer comments received and commented below on each of the points raised by the referees (printed in italics).

Reviewer 1:

In this manuscript, da Silva et al. developed biodegradable PLA/PBAT-based filaments for fertilizer release for agricultural applications. Structural, morphological, and thermal properties and the NPK releasing were evaluated. While the novelty of the entire work is insufficient and the research contents are not enough. In particular, the PLA/PBAT-based filaments proposed by the authors are quite common and cannot contribute to the progress of high-performance nanocomposite filaments. Therefore, I think the current paper is not suitable for the Materials.

Response:

Thank you for your review. We, unpretentiously, believe that the proposed biodegradable fertilizer-adsorbed filaments for employing in 3D printing to create complex structures able to controlled release fertilizers is novel, and simple to implement. Studies reported with this objective are absolutely arid in the literature and, additionally, the general idea of ​​the work can encourage further innovative works. We agree that PLA/PBAT-based filaments are quite common but we understand that adopting a blend that is relatively used in 3d printing as a starting matrix, for the formation of a composite with slow release capacity, would be more prudent to achieve the objective of producing the filament with NPK fertilizer and evaluate its potential for slow release. This potential proved to be promising as can be seen in the reported results.   

We believe that we had addressed all the concerns/questions raised by the reviewers and hope that the manuscript is now acceptable to Materials.

Yours sincerely

Reviewer 2 Report

The followings are the recommendations for the research paper entitled, "Development of biodegradable PLA/PBAT-based filaments for  fertilizer release for agricultural applications":

1. Author should include the final take-home message in the last line of the abstract.

2. L36: In addition to references no. 1 and 2, authors are advised to include the following references which reflects the fertilizer scenario of the south Aisa: https://doi.org/10.3390/agronomy11071306

https://doi.org/10.3390/agronomy11091756

3. After thoroughly reading the introduction part, I am not sure about the research gap of the study. Authors must include it clearly in the last para of the introduction part.

4. Some references are old. Authors are advised to improve the reference section in the Introduction part.

5. Methodology section is well elaborated. my only concern is the Characterization part (sub section 2.4). The author must illustrate the methodology, especially, how the TGA was done.

6. There is no clear-cut mention of the duration of the study. In which year, this study was conducted?

7. Results & Discussion are written nicely and illustrated with suitable graphs. However, the release kinetics should be more elaborative referring the figure 5.

8.  Conclusion must be rewritten elaborating the key results and possible recommendations and a line of future thrust of the study.

Author Response

Dear Editors,

Thank you for the opportunity to revise our manuscript "Development of biodegradable PLA/PBAT-based filaments for fertilizer release for agricultural applications". We have revised the manuscript in line with the reviewer comments received and commented below on each of the points raised by the referees (printed in italics).

Reviewer 2:

The followings are the recommendations for the research paper entitled, "Development of biodegradable PLA/PBAT-based filaments for fertilizer release for agricultural applications":

  1. Author should include the final take-home message in the last line of the abstract.

Response:

Thank you for your review. We agree with the Reviewer and have added a final take-home message (lines 29, 30). Due to word limitation, words in the original abstract were suppressed (lines 21, 28).

  1. L36: In addition to references no. 1 and 2, authors are advised to include the following references which reflects the fertilizer scenario of the south Aisa: https://doi.org/10.3390/agronomy11071306 https://doi.org/10.3390/agronomy11091756

Response:

We agree with the Reviewer and have added the indicated references (lines 38-45). It was necessary to add a background to integrate the new references (lines 38-45).

  1. After thoroughly reading the introduction part, I am not sure about the research gap of the study. Authors must include it clearly in the last para of the introduction part.

Response:

We agree with the Reviewer and have revised the last paragraph of the introduction (lines 95-98).

  1. Some references are old. Authors are advised to improve the reference section in the Introduction part.

Response:

We agree with the Reviewer and have added the new references (all the reference updates are highlighted in the manuscript).

  1. Methodology section is well elaborated. my only concern is the Characterization part (sub section 2.4). The author must illustrate the methodology, especially, how the TGA was done.

Response:

We partially agree with the Reviewer and have accepted the suggestion to improve methodology with illustrations. In our view, the characterizations carried out at work are common in materials research (SEM, XRD and TGA), so, given the prior work and space limitations, we chose not to devote specific illustration for TGA, but we took this suggestion into consideration and have added a schematic illustration of preparation of composites, production of filaments and 3D printing of the samples (line 124).

  1. There is no clear-cut mention of the duration of the study. In which year, this study was conducted?

Response:

The study was carried out over the years 2021 and 2022.

  1. Results & Discussion are written nicely and illustrated with suitable graphs. However, the release kinetics should be more elaborative referring the figure 5.

Response:

Thank you for this suggestion. In response to this comment, we have thus extended this discussion section referring the figure 5 (lines 334, 345).

  1. Conclusion must be rewritten elaborating the key results and possible recommendations and a line of future thrust of the study.

Response:

We agree with the Reviewer and have added the following sentences in the Conclusion:

“In this study, we produce filaments of PLA/PBAT and NPK fertilizer adsorbed on organophilized bentonite intended for application in the prototyping of biodegradable agricultural artifacts in 3D printing, using FDM technique” (lines 347-349).

“Especially, Ecov/6Bent/3NPK has the greatest potential for release control, with an aver-age cumulative release rate of 81.85% after 120 hours.” (lines 355-357).

“Finally, the filaments produced have the potential for slow or controlled release of ferti-lizer and are very promising to be applied in FDM in innovative planting techniques that can reduce the release of active compounds into the environment and mitigate environmental impacts. For future studies, compare dif-ferent types of organobentonite can be considered in order to evaluate the influence of surfactant (ionic and nonionic) on the adsorption of chemical species from fertilizers and on the release performance. A continuous process based on a twin screw extruder can also be considered in order to simplify filament production and future research will be pursued to determine the ecotoxicity of nanocomposites.” (lines 357-365).

We believe that we had addressed all the concerns/questions raised by the reviewers and hope that the manuscript is now acceptable to Materials.

Yours sincerely

Reviewer 3 Report

In this work, PLA/PBAT and NPK fertilizer filaments are produced using the fused deposition modeling (FDM) method and adsorbed over organophilized bentonite for usage in the 3D printing prototype of biodegradable agricultural artifacts. The use of regulated or slow release fertilizers (FLC) stands out among the most recent techniques researched to reduce fertilizer pollution. I agree to have this manuscript published in its current format.

The primary issue raised by the study concerned the production of PLA/PBAT and NPK fertilizer-adsorbed filaments on organophilized bentonite for use in the 3D printing of prototypes of biodegradable agricultural artifacts. I believe the concept of employing a 3D printer to create thin-walled tubular prismatic specimens as controlled or slow release fertilizers (FLC) is novel, affordable, and simple to implement. This technique may protect the environment from pollutants such as leaching of oxidized nutrients that contaminate aquifers and volatilization losses that produce air pollution. I think the paper is well-written, understandable, and simple to read. Additionally, the results and conclusion that are in conformity with the proposed reasons and supporting data also contain the claim that the proposed fertilizer-adsorbed filaments on organophilized bentonite releases slower than ten times of the traditional fertilizer.

Author Response

Dear Editors,

Thank you for the opportunity to revise our manuscript "Development of biodegradable PLA/PBAT-based filaments for fertilizer release for agricultural applications". We have revised the manuscript in line with the reviewer comments received and commented below on each of the points raised by the referees (printed in italics).

Reviewer 3:

In this work, PLA/PBAT and NPK fertilizer filaments are produced using the fused deposition modeling (FDM) method and adsorbed over organophilized bentonite for usage in the 3D printing prototype of biodegradable agricultural artifacts. The use of regulated or slow release fertilizers (FLC) stands out among the most recent techniques researched to reduce fertilizer pollution. I agree to have this manuscript published in its current format.

The primary issue raised by the study concerned the production of PLA/PBAT and NPK fertilizer-adsorbed filaments on organophilized bentonite for use in the 3D printing of prototypes of biodegradable agricultural artifacts. I believe the concept of employing a 3D printer to create thin-walled tubular prismatic specimens as controlled or slow release fertilizers (FLC) is novel, affordable, and simple to implement. This technique may protect the environment from pollutants such as leaching of oxidized nutrients that contaminate aquifers and volatilization losses that produce air pollution. I think the paper is well-written, understandable, and simple to read. Additionally, the results and conclusion that are in conformity with the proposed reasons and supporting data also contain the claim that the proposed fertilizer-adsorbed filaments on organophilized bentonite releases slower than ten times of the traditional fertilizer.

Response:  

We would like to thank the Reviewer for this very positive evaluation and for the comments.

We believe that we had addressed all the concerns/questions raised by the reviewers and hope that the manuscript is now acceptable to Materials.

Yours sincerely

Round 2

Reviewer 1 Report

Manuscript has been improved agree with Reviewers comments and suggestions. In my opinion manuscript can be publish in current form.

Author Response

Dear Editors,

Thank you for the opportunity to revise our manuscript "Development of biodegradable PLA/PBAT-based filaments for fertilizer release for agricultural applications". We have revised the manuscript in line with the comments received and commented below on each of the points raised by the Reviewer (printed in italics):

  • Some minor modifications should be made: release instead of re-lease, analysis instead of anal-ysis Cloisite instead of Closite.

 Response: Corrected (lines 23, 24, 25).

  • Some chemical formula are not written correctly.

 Response: Corrected (lines 105, 106).

  • The following sentence in really unclear, please explain:"These authors observed by XRD that the incorporation of Zinc did not cause phase changes in PBAT in systems with up to 3% Zn".

 Response: We thank the reviewer for this comment. The comparative choice of literature was not adequate. Although both studies use PBAT and fertilizers, the processing adopted and the type of fertilizer are very different, implying an inappropriate sentence. Therefore, we believe that it is more appropriate to withdraw this sentence.

  • Fig 5: add data of release for free NPK.

 Response: Data for free NPK has been added (line 306).

  • The following sentence is not correct:"For future studies, compare different types of organobentonite can be considered in order to evaluate the influence of surfactant..." replace compare by to compare.

 Response:

Corrected (line 355).

Other minor modifications were made and are highlighted in the manuscript.

We believe that we had addressed all the concerns/questions raised and hope that the manuscript is now acceptable to Materials.

Yours sincerely

Thyago Camelo Pereira da Silva, Allef Gabriel da Silva Fortes, Iago Rodrigues de Abreu, Laura Hecker de Carvalho, Yeda Medeiros Bastos de Almeida, Tatianny Soares Alves and Renata Barbosa.
